# A Seasonal Study of Koi Herpesvirus and Koi Sleepy Disease Outbreaks in the United Kingdom in 2018 Using a Pond-Side Test

**DOI:** 10.3390/ani11020459

**Published:** 2021-02-09

**Authors:** Irene Cano, John Worswick, Brian Mulhearn, David Stone, Gareth Wood, Jacqueline Savage, Richard Paley

**Affiliations:** International Centre of Excellence for Aquatic Animal Health, Cefas Weymouth Laboratory, Weymouth, Dorset DT4 8UB, UK; john.worswick@cefas.co.uk (J.W.); brian.mulhearn@icloud.com (B.M.); david.stone@cefas.co.uk (D.S.); gareth.wood@cefas.co.uk (G.W.); jacqueline.savage@cefas.co.uk (J.S.); richard.paley@cefas.co.uk (R.P.)

**Keywords:** cyprinid herpesvirus, carp edema virus, common carp, fluorescence real-time loop-mediated isothermal amplification, LAMP, point of care test, skin swab, disease control

## Abstract

**Simple Summary:**

Cyprinid herpesvirus (CyHV)-3 and carp edema virus (CEV), the causative agents of koi herpesvirus disease and koi sleepy disease, respectively, are emerging DNA viruses infecting koi and common carp. Similarities in their clinical presentation present difficulties for its on-site identification based on gross pathology. Fluorescence real-time loop-mediated isothermal amplification (LAMP) assays for detecting CyHV-3 and CEV DNA were designed to use border inspection posts and local testing by national authorities for outbreak control. The limit of these tests’ detection (10^2^ and 10^3^ viral copies for CyHV-3 and CEV, respectively) allows for the amplification of viral DNA in clinical samples in less than 20 min. The assays’ field performance was tested with 63 common carp mucus swabs taken during disease investigations in 2018, and the results validated with the reference laboratory analysis. Overall, the good performance, ease of use, and cost-effectiveness of these tests make them good candidates for a point of care test. However, further work is required to incorporate reliable internal controls and improve the sensitivity of these tests’ asymptomatic testing.

**Abstract:**

Fluorescence real-time LAMP assays were designed for the *orf43* gene of CyHV-3 European genotype and the *p4a* gene of the CEV genogroup I. A third LAMP assay to detect the *ef1a* gene of the host common carp was designed as an internal control. The limit of detection was 10^2^ and 10^3^ viral copies under 25 min for CyHV-3 and CEV, respectively. The specificity of the CyHV-3 LAMP assay was 95.6% of 72 fish herpesviruses tested. Sixty-three non-lethal common carp mucus swabs were collected across 16 sites during disease investigations. DNA extractions were performed in under 10 min using the QuickExtract™ digestion buffer. The LAMP amplification of CyHV-3 DNA in mucus swabs from clinical cases was detected from 4 to 13 min in 13 sites, while a co-infection of CyHV-3 and CEV was confirmed by LAMP in a single site. The LAMP results agreed with the results of the reference laboratory. The common carp *ef1a* was amplified only in 61% of the mucus swabs collected, preventing its use as a robust internal control to distinguish false negatives from invalid tests. After further optimization, these tests could be implemented for border inspection posts surveillance and decentralizing testing during disease outbreaks.

## 1. Introduction

Common carp (*Cyprinus carpio* L.) is one of the most cultured freshwater fish species worldwide, with global production estimated at 4.1 million tons in 2017, about 8.3% of the total global inland aquaculture production [1]. Common carp is also a highly traded species, with approximately 99,000 tons of live, fresh/chilled filleted, or frozen carp products traded annually [1]. Diseases cause significant losses in cyprinid fish farming. Due to the demand for common carp commercial trade, there is a great risk of introducing cyprinid viral diseases to new areas [2]. Koi herpesvirus disease (KHVD), koi sleepy disease (KSD), and carp pox disease (CPD) are currently among the most significant viral diseases for carp production [2]. 

Cyprinid herpesviruses (CyHVs) are large enveloped double-stranded (ds)DNA viruses belonging to the genus *Cyprinivirus* within the family *Alloherpesviridae* [3]. CyHV-1, or carp pox virus, is the causative agent of CPD. Susceptible species are common carp, crucian carp (*Carassius carassius* L.), and orfe (*Leuciscus idus* L.). CyHV-1 infection in fry manifests as an acute systemic lethal disease. In older fish, CPD is characterized by a recurring proliferative skin disease usually linked to seasonal periods of lower water temperature, typically between 9–16 °C [4]. CyHV-2, the causative agent of herpesviral hematopoietic necrosis (HVHN), typically infects goldfish (*Carassius auratus* L.), but also crucian carp and Gibel carp (*C. auratus gibelio* L.) at water temperatures of 20–25 °C [5,6].

CyHV-3, or koi herpesvirus (KHV), is the causative agent of KHVD, a notifiable disease to the World Organization for Animal Health from 2006 (OIE, 2019). Since its first detection in the 1990s, KHVD has spread to most regions worldwide due to the global fish trade [7]. Typically, disease outbreaks occur between 17–28 °C in common and koi carp [8]. Clinical signs are characterized by white patches and hemorrhages on the skin leading to severe lesions on the epidermis, lethargy, lack of appetite, enophthalmos (sunken eyes), enlargement of the spleen and kidney, and gill necrosis [9,10]. CyHV-3 establishes latent infection in white cells [11], reactivation, and shedding (asymptomatic or symptomatically) after heat stress has been demonstrated [12,13]. Based on phylogenetic analysis, three genetic lineages of CyHV-3 described as European, Asian, and Intermediate have been described [14,15]. Moreover, CyHV-3 variants sharing nucleotide identities of 95–98% with CyHV-3 strains have also been identified, usually infecting asymptomatic hosts [16].

KSD is an emerging disease caused by carp edema virus (CEV), a large, enveloped dsDNA virus belonging to the *Poxviridae* family. First detected in Japan in the 1970s, it has expanded fast in Europe and North America, and more recently in China and India [17], likely introduced through the international carp trade. In the United Kingdom (UK), it was first detected in 2009 in imported koi, then subsequently in common carp in fisheries in 2012 and has been detected in archive material as far back as 1998 [18]. Clinical signs can include lethargy and unresponsiveness, anorexia, erosive or hemorrhagic skin lesions, enophthalmos, pale swollen gills, and gill necrosis. A wide temperature range from 6 to 24 °C has been recorded during the reported cases, although most of the outbreaks fall within 19 to 24 °C [19]. Phylogenetic analysis based on the partial *core protein 4a* gene (*p4a*) classified CEV into two genogroups: I and II, the latest one split onto clades IIa and IIb, being genogroup I the most abundant infecting common carp in the United Kingdom (UK) [20].

In the UK, disease outbreaks during the spring session often offer overlapping permissive temperatures for CyHV-1, CyHV-2, CyHV-3, and CEV infection. Co-infections of CyHV-3 and CEV [21,22] have also been reported. Similarities in the clinical presentation of KHVD, KSD, and CPD, mostly related to gill necrosis and skin lesions, can present difficulties for its on-site identification based on gross pathology. To further complicate a local veterinary diagnosis, other carp viruses that can cause gill necrosis and skin and gill petechial hemorrhages are the common carp paramyxovirus (CCPV), an enveloped negative-sense single-stranded (ss)RNA virus [6,23], and the OIE listed spring viremia of carp (SVC). Outbreaks of SVC virus (SVCV), a negative-sense ssRNA rhabdovirus, are largely influenced by water temperatures falling between 10–17 °C [24], potentially outside of the susceptible temperatures for the onset of KHVD. From 2010, the UK was recognized as free of the SVC following an exhaustive control and eradication program for SVCV initiated in 2005 [25].

Since 2007, KHVD is notifiable in the UK via the European Directive 2006/88/EC [26]. Official inspections conducted by the Fish Health Inspectorate (FHI) on suspected outbreaks of KHVD are followed by restrictions on fish movement, statutory laboratory confirmation of CyHV-3 from gill tissue samples, and subsequent monitoring and control programs. CyHV-3 diagnostics are conducted by viral isolation on susceptible cell lines and conventional PCR amplification and sequencing targeting either the viral *DNA polymerase* gene with cyprinid herpesvirus generic primers [16], the *thymidine kinase* (*tk*) gene with CyHV-3 specific primers [27], and the viral *orf 90* amplified by a Taqman real-time PCR assay [28]. In parallel, the presence of CEV on the same gill samples is routinely analyzed. CEV *in vitro* cultivation has not been successful so far, and its diagnostics is based on PCR detection of the core protein *p4a* gene and sequencing of the PCR amplicon [20].

Thus, although outbreaks in fisheries can be managed by close monitoring and eradication programs to avoid further pathogen introduction throughout the international trade, the FHI in England and Wales conduct random surveillance screening on imported live fish arriving at border inspection posts (BIPs). BIP surveillance in cyprinid species is aimed to detect both listed (SVCV and CyHV-3), emerging (i.e., CEV), as wells as other non-listed and exotic pathogens (i.e., Chinook Salmon Bafinivirus (CSBV)) [29].

Recently, numerous isothermal assays have been designed based on their potential for user-friendly rapid diagnostic tests of the main aquaculture pathogens. More popular detection methods use either helicase dependent amplification (HDA) [30], recombinase polymerase amplification (RPA) [31], or loop-mediated isothermal amplification (LAMP) [32,33,34] chemistry, among others. Those platforms, coupled with field sample preparation consisting of fast DNA/RNA extraction methods based on either digestion buffers [34], lateral flow devices (LFD) [35], magnetic solid-phase reversible immobilization (SPRI) [36], or microfluidics cartridges [37] among others, offer great potential for its use as point-of-care tests (POCT).

Even though fast diagnostic assays with potential for use as POCT for CyHV-3 and CEV have been already designed [31,38,39,40,41,42], very little is known about their current field application. The present study aimed to evaluate a novel fluorescence real-time LAMP assay for the on-site discrimination of CyHVs and CEV in non-lethal mucus samples of common carp clinically infected and sampled during disease outbreaks investigated in England and Wales during spring and summer of 2018. Pond-side results were blind compared with the laboratory-based statutory diagnostics. Its robustness and potential applicability as POCT is compared with other tests published. The feasibility of using POCT in the field and BIPs are discussed.

## 2. Materials and Methods

All the animals used in this study were sampled as a result of official disease investigations. These animals were not subjected to a regulated procedure. Fish were euthanized humanely according to the UK Home Office procedures in compliance with the Animals (Scientific Procedures) Act 1986 Amendment Regulations 2012.

### 2.1. DNA Control and Recombinant Plasmids

A European strain of CyHV-3, isolate K250, was propagated in the common carp brain (CCB) derived cell line (ECACC 10072802) at 20 °C in EMEM media supplemented with 2 mM Glutamine, 1% non-essential amino acids, 2% Fetal Bovine Serum (FBS) and 10 mM HEPES (Sigma-Aldrich, Gillingham, UK) [43]. The supernatant of CCB cells showing cytopathic effects was clarified by centrifugation at 4000× *g* for 15 min to pellet the cell debris. Viral nucleocapsid was then extracted from the clarified supernatant using the EZ1 Virus Mini Kit and the EZ1 extraction robot (Qiagen, Manchester, UK) following the manufacturer’s instructions as a positive control on the LAMP reactions.

A fragment of 1450 bp of the CyHV-3 *orf90* gene was cloned into the pGem-T Easy plasmid vector (Promega, Southampton, UK) as described before [44]. The purified DNA plasmid was then used to generate a standard curve for the qPCR quantification. The template (dsDNA) copy number was calculated using a QuantiFluor dsDNA kit in a Quantus fluorimeter (Promega, Southampton, UK), and a dilution series, from 10⁠^7^ to 1 copy, was generated.

For the CEV LAMP assay, a fragment of 528 bp of the CEV *p4a* gene was amplified using the set of primers CEV ForB and RevJ [18] and cloned as described above. The purified plasmid DNA was used as a positive control for the CEV qPCR and the LAMP assay.

### 2.2. LAMP Primers Design

A LAMP assay for detecting CyHV-3 was designed in a conserved region of the *orf43* gene of the CyHV-3 KHV-U strain [14] belonging to the European genotype (GenBank accession number NC_009127). Primers were designed using the LAMP Designer 1.10 program (Premier Biosoft International), consisting of two outer primers (F3 and B3), two inner primers (FIP and BIP), and two loop primers (Loop-F and Loop-B) (Table 1), targeting a region of 129 bp (Figure 1A).

A second LAMP assay for detecting CEV-I was designed in a conserved region of the *p4a* gene of the CEV Q030 strain belonging to the genogroup I (GenBank KX254013.1). Primers were designed as described above, targeting a 224 bp product (Table 1, Figure 1B).

A third LAMP assay, used as an internal control, was designed to amplify a fragment of 143 bp of the common carp *elongation factor 1 alpha* (*ef1a*) gene (GenBank no. AF485331.1) (Table 1, Figure 1C).

### 2.3. LAMP Assay Temperature Optimization and Analytical Sensitivity

The isothermal reaction temperature was optimized for detection of CyHV-3 using a block gradient from 60 to 67 °C at a 1 °C interval followed by an annealing step of 98–80 °C, ramping at 0.05 °C per second. For its field application, the CEV and common carp *ef1a* LAMP assays were run at the same temperature chosen for the CyHV-3 test.

LAMP reactions consisted of 15 μL of the fast isothermal master mix (ISO-004, OptiGene), 5 pmol of each primer F3 and B3, 10 pmol of each Loop-F and Loop-R, 20 pmol of FIP and BIP, and either 5 μL of the extracted DNA or DNA plasmid control and nuclease-free water to a final volume of 25 μL.

Isothermal amplification was performed either in a Genie^®^ vII or vIII system (OptiGene) for real-time monitoring of the LAMP amplification. The amplification ratio was measured as the change of fluorescence over time, expressed as the time of positivity (*Tp*, mm:ss). *Tp* and the amplicon annealing temperature were visualized using a Genie^®^ vII or vIII software (OptiGene).

Then 10-fold serial dilutions of recombinant plasmids were used as described above to determine the limit of detection (LOD) of the CyHV-3 and the CEV LAMP assays. Linear regression analysis between the number of copies and *Tp* was performed from three different independent assays.

### 2.4. Test Specificity 

The specificity of the primers was tested in silico against representatives of the three CyHV-3 genotypes. A multiple sequence alignment (MegAlign v7.0.21; Lasergene, DNASTAR) was conducted against the *orf43* gene of the European genotype, strain I (MG925489.1 [45]) and FL (MG925487.1 [46]); Asian genotype, strains T (MG925491.1 [47]) and M3 (MG925490.1 [45]); and Intermediate genotype, strains GZ11-SC (MG925488.1 [48]) and GZ11 (KJ627438.1 [48]). 

The nucleotide identity of the LAMP probing region was compared with the *orf43* gene of the relatives CyHV-1, strain NG-J1 (NC_019491.1), and CyHV-2, strain SY (KT387800.1), and with an *orf43* ortholog gene of an eel herpesvirus, the *orf19* AngHV-1 (NC_013668.3) [49,50]. The CyHV-3 LAMP assay was run with archived gill homogenates that tested positive for CyHV-1 (28 DNA samples), CYHV-2 (7 samples), AngHV-1 (2 samples), a CyHV-3 variant (30 samples), and with uninfected common carp gill tissues (47 samples).

For the CEV LAMP assay, a multiple sequence alignment against the CEV *p4a* gene of representatives of the genogroup I, strains Q229_2.2 (KX254019.1 [20]) and Q030_1.2 (KX254013.1 [20]); genogroup IIa, strains 687-2014 (KX254000.1 [20]) and 274-2014 (KX254003.1 [20]; and genogroup IIb, strains 548-2014 (KX254006.1 [20]) and 396-2013 (KX254005.1 [20]) were constructed. The primers’ specificity was also tested *in silico* against the *p4a* gene of another member of the family *Poxviridae*, the salmon gill pox virus (NC_027707.1) [51]. The assay was run with a representative of CyHV-1, CyHV-2, and CyHV-3 and uninfected common carp gill tissues (30 samples).

### 2.5. Pathogen Identification by Standard PCR and Sequencing, and Taqman qPCR

The presence of CyHV-3 DNA was confirmed either by Taqman qPCR targeting a fragment of the viral *orf90* gene [28] or by a PCR and nested-PCR targeting the viral *tk* gene [27]. The presence of CEV *p4a* gene DNA was analyzed using a Taqman PCR assay described previously [18]. In addition, the presence of other CyHVs was confirmed by PCR and nested PCR using a generic set of primers (CyHV generic PCR, Table 1), which target the viral *polymerase* gene and have been shown to amplify all three CyHVs and eel herpesvirus (species type anguillid herpesvirus 1 (AngHV-1)) [16]. PCR products were sequenced using the BigDye^®^ Terminator v3.1 Cycle Sequencing Kit (Applied Biosystems, Warrington, UK) following the manufacturer’s recommendation. Sequence analysis was performed with the ABI 3500 xl Genetic analyzer (Applied Biosystems, Warrington, UK) and CLC DNA analysis software (Qiagen, Manchester, UK). Nucleotide similarity was determined by BLASTn [52].

### 2.6. Testing CyHV-3 LAMP Assay with Archived DNA Samples

Two hundred and four DNA samples were obtained from the Cefas archive sample collection, consisting of 85 CyHV-3, 72 other fish HVs, and 47 negative controls, and blind tested in duplicate using the CyHV-3 LAMP assay and compared with the standard diagnostic tests described in Section 2.5. DNA was extracted from gill tissues using either DNAzol (Invitrogen, Inchinnan, UK) or EZ1 Virus Mini Kit v2.0 extraction cartridges in a BioRobot EZ1 (Qiagen, Manchester, UK). The samples were predominantly taken from carp (koi, common, mirror, ghost, and crucian) but also included samples from goldfish, rudd (*Scardinius erythrophthalmus* L.), roach (*Rutilus rutilus* L.), tench (*Tinca tinca* L.), and eel (*Anguilla anguilla* L.).

### 2.7. Application of CyHV-3 and CEV LAMP Tests in Non-Lethal Mucus Swabs Samples Taken during Disease Investigations

A total of 63 mucus swabs were taken from 16 sites distributed in England and Wales during disease investigations that occurred in spring and summer 2018. Examples of the presentation of clinical signs can be found in Appendix A. During the disease investigations, gill tissues of a minimum of 30 fish per site were collected for statutory diagnostics in the laboratory. In parallel, from some of those specimens, mucus swabs, either from the gill and/or skin, were taken to evaluate the POCT. The number of swabs collected per site varied from 1 to 10, each swab corresponding to a different fish. The inconsistency in the number of swabs taken per site was due to the practicability of taking samples during the official investigations, which varied from site to site. 

Isohelix DNA Buccal Swabs (Sigma-Aldrich, Gillingham, UK) were used to take non-lethal mucus samples of common carp gills showing clinical signs of disease. In parallel, gill tissues were dissected for statutory diagnostics as part of a formal FHI investigation. 

As a POC DNA extraction method from the skin and gill swabs, the QuickExtract™ DNA extraction solution (Cambio, Cambridge, UK) was used following the manufacturer’s recommendations. Swabs were placed in 500 µL of the QuickExtract™ buffer and incubated for 6 min at 65 °C followed by 2 min incubation at 98 °C conducted on the same Genie^®^ vII or vIII equipment used subsequently in the LAMP reactions. Two LAMP reactions (CyHV-3 and CEV) were run in parallel in duplicate wells for each sample. In the same LAMP strips, negative (containing extraction buffer) and positive (containing 10^3^ copies of either CyHV-3 or CEV recombinant plasmid DNA) controls were added. The common carp *ef1a* LAMP assay was used as an internal control for inhibitors’ presence on the DNA extraction. 

LAMP results were compared with those conducted in the laboratory. Briefly, in the laboratory, viral nucleic acid was extracted from 100 µL of clarified gill tissue homogenates (1/10 in transport media (Glasgow’s MEM, supplemented with 10% fetal bovine serum, 200 IU mL^−1^ penicillin, 200 μg mL^−1^ streptomycin, and 2 mM L-glutamine)) using an EZ1 Virus Mini Kit in an EZ1 extraction robot (Qiagen, Manchester, UK) and the presence of CyHV1, CyHV3, and CEV DNA scrutinized by Taqman qPCR as described above. 

## 3. Results

### 3.1. Fluorescence Real-Time LAMP Test Optimization and Limit of Detection

The detection of CyHV-3 control DNA ranged from 7:10 mm:ss at 63, 64 and 65 °C (faster detection) to 7:55 mm:ss at 67 °C (slower detection). 64 °C was then selected as an optimal reaction temperature for the CyHV-3 LAMP assay (Figure 2A). The LAMP products’ annealing curve, using either the recombinant plasmid or extracted DNA, showed a single peak in the range of 89.5–90.5 °C for CyHV-3 (Figure 2B,D). As an average of three independent runs, the CyHV-3 LAMP assay detected 10^6^ copies of the recombinant plasmid under 9 min and 10^3^ copies under 16 min. Detection of 10^2^ copies of the target gene required a longer time (*Tp* ~24:00 mm:ss). Ten copies of the recombinant plasmid were detected occasionally (1 out of 3 runs), while one copy was not detected (Figure 2C). The limit of detection (LOD) of the CyHV-3 assay was then established as 10^2^ copies in a run lasting at least 25 min. 

For standardization and multiplexing (in the same strip) purposes, the CEV and common carp *ef1a* LAMP assays were run at the same temperature as the CyHV-3 test. The CEV LAMP assay detected 10^6^ copies of the recombinant plasmid under 13 min and 10^3^ copies under 18 min. Detection of 10^2^ copies showed an unspecific amplicon at ~52 min with an annealing temperature different than the expected at 82 °C for CEV (Figure 3A,B). The 10 and 1 copy of the recombinant plasmid were not detected. The LOD of the CEV assay was then established in 10^3^ copies in a 20 min run.

For both tests, linear regression analysis between the detected copy numbers and the *Tp* showed a good correlation (Pearson’s r = −0.97 and −0.96 for CyHV-3 and CEV LAMP assays, respectively) (Figure 4).

### 3.2. Specificity

The CEV LAMP assay showed a nucleotide identity in the LAMP probing region of 99% for isolates of the genogroup I; 94–100% with isolates of genogroup IIa; and 92–94% with isolates of the genogroup IIb. Blast analysis with its closest relative within the family *Poxviridae*, the salmon gill poxvirus, showed no binding of the primers in the probing region.

The CEV fluorescence LAMP test did not produce a fluorescent signal when using DNA extracted from uninfected common carp (30 samples) or with control DNA (*n* = 30) of CyHV-1, CyHV-2, and CyHV-3 (*n* = 3).

Nucleotide comparisons of the *orf43* gene of the selected CyHV-3 European, Asian, and Intermediate strains showed a 100% similarity in the LAMP probing region. There were not significant similarities among the predicted CyHV-3 LAMP product and the genome of the isolates NG-J1 (a CyHV-1), SY (a CyHV-2), or the AngHV-1 ortholog ORF19. 

The CyHV-3 LAMP assay was tested with positive DNA extracted in duplicate from 85 archived gill tissue samples from common carp and compared with PCR (*tk* gene) and/or Taqman qPCR (*orf90* gene) assays (Table 2). A test was counted positive when at least one of the duplicate reactions was positive. Of those 85 confirmed positives, only 76 samples tested positive in the LAMP assay. The nine samples that did not produce an amplification signal with the LAMP assay previously testing positive in the second round of PCR (nested PCR) only and 3 of the samples tested positive by nested PCR in only one of the two duplicate extractions. Two of the nine samples were not detected by Taqman qPCR. Of the seven detected by Taqman qPCR, there was amplification in both reactions, but the mean copy number derived from the standard curve indicated between 5.4 copies and a maximum of 39.6 copies in the reaction below the estimated LOD of the LAMP reaction (~100 copies). 

A further 119 samples were tested; 30 corresponded to a CyHV-3 variant, 28 were positive for CyHV-1, 10 were CyHV-2, 2 were AngHV, there were two uncharacterized herpesvirus (HV), and the last 47 samples corresponded to an uninfected fish. The LAMP test did not cross-react with samples from the population harboring the CyHV-3 variant DNA. However, the concentration of the CyHV-3 variant in this population was low (only detected by the generic CyHV nested-PCR). Of the 28 CyHV-1 samples, three of them gave a LAMP amplification. Of those amplifications, two samples showed a non-specific LAMP product with a melting temperature different from the predicted for CyHV-3 (M068 2.1). While the third sample (M065 2.1), a CyHV 1-like showing a 98% homology to carp pox virus in the polymerase gene, one of the duplicate gave a LAMP product which a melting temperature not discernable from CyHV-3 (false positive) (Figure 5). Of the 10 goldfish herpesvirus (CyHV-2) samples, three of them resulted in amplification in the CyHV-3 LAMP assay (Figure 5). Of those amplifications, a sample (L112 2.1, Figure 5C,D) showed a second peak in the range of CyHV-3 melting temperature for one of the duplicates. This sample tested positive for CyHV-3 by Taqman qPCR at a very low copy number; thus, the test was interpreted as correct. One of the duplicates of the CyHV-2 sample from goldfish (M084-2.2, Figure 5E,F) also showed LAMP amplification of low intensity, but the melt analysis cannot distinguish the product from CyHV-3. Another CyHV-2 sample (M084-2.1, Figure 5A,B) also gave a LAMP product in both duplicates within the CyHV-3 melting temperature.

Two uncharacterized herpesviruses from mirror carp, showing the highest nucleotide homology to sturgeon and catfish herpesviruses (74% and 62%, respectively), were also tested. One resulted in LAMP amplification in one of the duplicates that were not different from CyHV-3 by melt analysis (data not shown).

As a summary, the CyHV-3 LAMP test gave a total of 10.5% of false negatives (76 correct detections out of 85), which corresponded with viral copy number lower than 100 in the test tube, and 9.5% (4 out of 42) false negatives showing the same melting temperature than the expected for CyHV-3. 

### 3.3. CyHV-3 and CEV LAMP Test of Non-Lethal Swabs of Clinically Infected Common Carp

The fast and dirty DNA extraction from mucus swabs using the POC method ranged from 178 to 882 ng µL^−1^ averaging 431 ± 190 ng µL^−1^. The internal control common carp *ef1a* gene was amplified from the mucus swabs only in 38 out of 63 swabs (60%). In those 38 swabs, the time for detecting the common carp *ef1a* gene varied from 01:45 to 15:30 mm:ss, averaging 07:50(±02:50) mm:ss. The amplified LAMP product’s melting temperature gave a single peak at 88.15 (± 0.14) °C (Figure 6A,B). 

The presence of the *orf43* gene of CyHV-3 was analyzed by a LAMP test in all the swabs collected. A site was considered CyHV-3 positive when at least one of the swabs tested positive. The percentage of positive swabs in each site varied from 100% (all swabs positives) to 0% (all negatives) (Table 3). The *Tp* in CyHV-3 LAMP positive swabs ranged from 04:45 to 13:03 mm:ss, averaging 07:50(±02:50) mm:ss with a single peak in the melting curve of 89.7(±0.15) °C (Figure 6C,D; Appendix A). 

When the same DNA extracted from the swabs were analyzed by Taqman qPCR, 4 out of 51 (7.8%) of the swabs that tested positive by LAMP resulted negative by qPCR, while 5 out of 6 (83%) of the swabs that were negative by LAMP resulted positive by qPCR. For those swabs LAMP negative but qPCR positive, the qPCR cycle threshold (CT) values were high (>35), indicating the presence of viral DNA below the LOD of the LAMP test. 

CEV DNA was detected only in one site (site C) in 2 out of the 5 swabs analyzed, and the recorded *Tp* was 19:30 and 20:00 mm:ss for each swab, respectively. The annealing curve gave a single peak of 82 °C (Figure 6E,F). Those positive CEV LAMP swabs were also CyHV-3 positive. The CyHV-3 and CEV co-infection in this site was later confirmed in the laboratory in gill homogenate samples. 

The POC LAMP tests using mucus swabs were interpreted as CyHV-3 positive in 13 out of the 16 sites, and the co-infection of CyHV-3 and CEV was identified in one of the 16 sites visited. The statutory diagnostic corroborated the LAMP results in gill homogenates and by Taqman qPCR confirming the presence of either CyHV-3 or CEV in those LAMP positive sites and the negativity of the other sites. 

## 4. Discussion

The diagnostic of veterinary notifiable diseases is conducted in accredited reference laboratories following the diagnostic tests recommended by the OIE manuals. Centralized testing assures uniformity of the testing conforming quality assurance systems but might also imply a delay in the results due to the samples’ shipment, which could affect immediate decision-making and disease control. Point of care tests offers the possibility of decentralizing testing. The tests can be beneficial in specific situations, such as surveillance programs in border inspection posts to advise on quarantine or movement restrictions in fish imports. They are also beneficial in local testing to stop the spread of infectious diseases once the disease has been confirmed in the reference laboratory to diagnose secondary cases, with the potential to decrease the costs associated with outbreak control (i.e., the unnecessary slaughter of uninfected animals) [53].

Up to date, numerous POCTs have been published to detect a wide range of fish and shellfish pathogens [32]. However, none of them are included in the OIE aquatic manual [54]. Currently, only one POCT, the Ag-LFD for the detection of foot and mouth disease virus (FMDV) [55], is included in the OIE terrestrial manual as an approved diagnostic test [56]. In order to incorporate a POC assay as a validated test in the OIE manuals, a rigorous validation is necessary, which might require ring testing involving national reference laboratories to confirm that the test is “fit to propose”, which includes validated data on performance, sensitivity, and specificity [57].

Several fast assays with the potential for its use as POCT have already been designed for the detection of CyHV-3, such as a colorimetric LAMP test [38]; two real-time turbidimeter LAMP assays [41,42]; an RPA assay designed to detect CyHV-3 latency on white blood cells [40]; a colorimetric gold nanoparticle-based hybridization assay [39]; and an LFD multiplex RPA assay, the last one a multiplex assay to detect CyHV-3 and CEV DNA [31]. However, there are no actual records of their depletion in the field or their incorporation into contingency plans. Thus, these tests are likely to remain underexploited. In the present study, novel fluorescence real-time LAMP assays for detecting CyHV-3 and CEV DNA were designed for a Genie^®^ platform. This equipment allows for the transmission of the data through Bluetooth and GPS, thus enabling a direct connection in real-time of the POCT results with the reference laboratory. 

The CEV LAMP primers were designed to amplify the *p4a* gene of CEV genogroup I, which is the most abundantly found infecting common carp in the United Kingdom (UK) [20]. However, isolates from genogroups IIa and IIb have also been reported within the UK [58]. A potential application of this assay is BIP screening of cyprinid imports to stop introducing infected animals; therefore, an assay able to detect all the genogroups was desired. Up to date, only CEV sequences of a small fragment of the *p4a* gene are available, thus preventing the assay’s design in a conserved region. In the present study, the CEV assay was only tested with isolates of genogroup I; thus, it remains unknown how this assay might perform with isolates of other genogroups. 

The CyHV-3 fluorescence real-time LAMP assay specificity was widely tested with a large number of archived samples harboring CyHVs DNA. None of the negative samples gave a positive result. However, despite blast analysis indicating the primers’ specificity to CyHV-3 strains, the test showed a low level of cross-reaction (4.4%) with other fish herpesviruses. Nucleotide identities might explain the cross-reaction with an uncharacterized herpesvirus in the probing region. However, there are not sequences available for this strain. Recently, sequencing studies are shown the presence of a variety of CyHV-3-like, called CyHV-3 variants, in common carp [16]. Those strains share 95–98% with CyHV-3 for the DNA polymerase and the major capsid protein. The present LAMP assay did not cross-react with samples harboring a CyHV-3 variant DNA. However, the amount of viral DNA in those samples likely was close to the LOD of the test. Thus, it remains unknown if this test could potentially cross-react with those strains at a higher concentration. 

The analytical sensitivity of the fluorescence real-time LAMP assays was similar to a standard PCR with the advantage of the visualization of the results under 20 min. Previous fast assays for the detection of CyHV-3 also reported LODs similar to a PCR [31,38,39,42] or a qPCR (~10 copies) [39,40] (Table 4). However, the advantage of the fluorescence real-time LAMP assay over those tests is that the visualization of the results is in real-time, which make this assay the fastest (i.e., clinical samples were detected at 07:50(±02:50) minutes) without the requirement of further pipetting (as in colorimetric reactions) or the use of a second device (as an LFD). 

To address if the LOD of the CyHV-3 POCT was fit to propose, the assay’s performance was tested in CyHV-3 positive archived samples. The LAMP test successfully identified the pathogen in 89.4% of the samples under 20 min. The viral copy number in the false negatives was less than 50 copies under the LOD of the LAMP test. Typically, in CyHV-3 symptomatic infections (viral productive phase), between 10^4^–10^9^ copies of the viral DNA can be detected in gill tissues (viral copies referenced to 10^6^ host cells); while in asymptomatic specimens (latent phase), the viral copy number in gills tissues drops below 100 [28]. Thus, the LOD of this CyHV-3 POCT allows for its use in clinically infected specimens, but it is not recommended for surveillance programs in asymptomatic populations where the virus might be present at a very low copy. Even though peripheral blood leukocytes can support CyHV-3 latency in asymptomatic carp [12,59], only 1% of the leukocytes carry a few copies of the virus [59]. Consequently, for the detection of CyVH-3 latent infections, only highly sensitive tests as nested-PCR, qPCR, and a highly sensitive RPA assay have been successful so far [11,40].

The host *ef1a* LAMP assay was run parallel to the CyHV-3 and CEV LAMP assays as an internal control to identify invalid tests. In gill swabs from Atlantic salmon, the host *ef1a* gene’s detection resulted in a reliable internal control [34]. However, this internal control did not perform as expected in mucus swabs from common carp. Currently, there are other options to test for the presence of inhibitors in the DNA extraction, for example, adding a known concentration of a synthetic DNA in the test sample [60]. The design of a reliable internal control is highly recommended for implementing these assays as a POCT to demonstrate the capability of identifying invalid tests. 

Of the 16 sites visited during disease investigations in 2018, the real-time LAMP analysis identified CyHV-3 in 12 sites and co-infection of CyHV-3 and CEV in a further site. Taqman qPCR of gill homogenates conducted in the reference laboratory confirmed the LAMP designation to be accurate in 100% of the sites. The test of a minimum of 3 mucus swabs per site in symptomatic animals was seen to be enough for the CyHV-3 detection in at least one swab. Previous studies have shown that clinically diseased common carp shed high loads of CyHV-3 [61]. However, there are no previous data that support CEV shedding from clinically infected animals. Although this emerging virus’s pathogenesis and transmission routes are still to be elucidated, it has been shown that CEV infects and causes lesions primarily in gill tissues [19]. The mucus layer in common carp is composed of water (~95%), glycoproteins (as mucins), and other substances (as cytokines, lysozyme, complement, and antibodies, among others) [62]. A gill swab not only removes mucus associated with the epidermis but also exfoliates the epidermis layer and potentially remove infected cells.

Interestingly, CyHV-3 and CEV co-infections associated with mortality have been reported recently [22,63]. However, due to clinical symptoms’ similarity, CEV infection might have been underreported in historical CyHV-3 outbreaks [21]. Both CyHV-3 and CEV were detected in mucus swabs, showing an active infection and viral shedding of both viruses. The role of CEV infection in the severity of the pathology and outbreaks requires further investigation. 

## 5. Conclusions

Overall, the good performance, ease of use, and cost-effectiveness of the CyHV-3 and CEV real-time LAMP assays with clinical field samples encourage incorporating these tests in contingency plans, and BIP controls to work alongside reference laboratories for disease control. However, further work is required to (1) incorporate reliable internal controls to distinguish invalid tests from false-negative results, (2) more reference genomes to allow for a better design of the primers to improve the specificity of the test (in the case of CyHV-3) and for the detection of other circulating genogroups (in the case of CEV), and (3) to improve the sensitivity of the tests below 100 copies of viral DNA to allow for asymptomatic testing.

## Figures and Tables

**Figure 1 animals-11-00459-f001:**
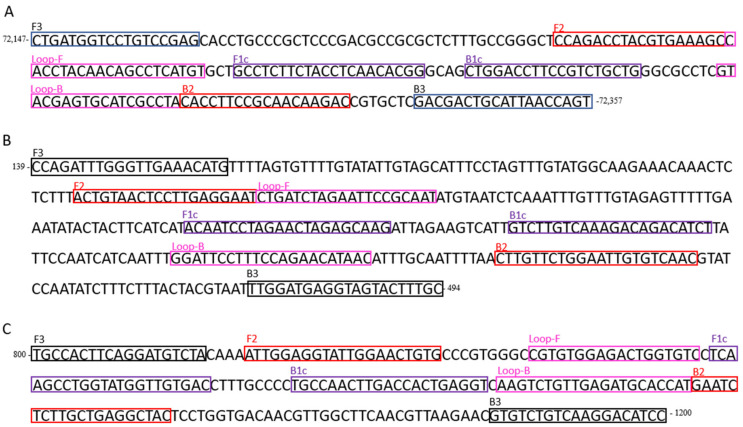
Primers position and probing region for the loop-mediated isothermal amplification (LAMP) of (**A**) Cyprinid herpesvirius-3 (*orf43* gene, NC_009127); (**B**) Carp Edema Virus (*p4a* gene, KX254013.1); and (**C**) common carp (*ef1a* gene, AF485331.1). Outer primers (F3 and B3) are boxed in blue; inner primers (FIP and BIP) are boxed in red and purple respectively; loop primers (Loop-F and Loop-B) are boxed in pink.

**Figure 2 animals-11-00459-f002:**
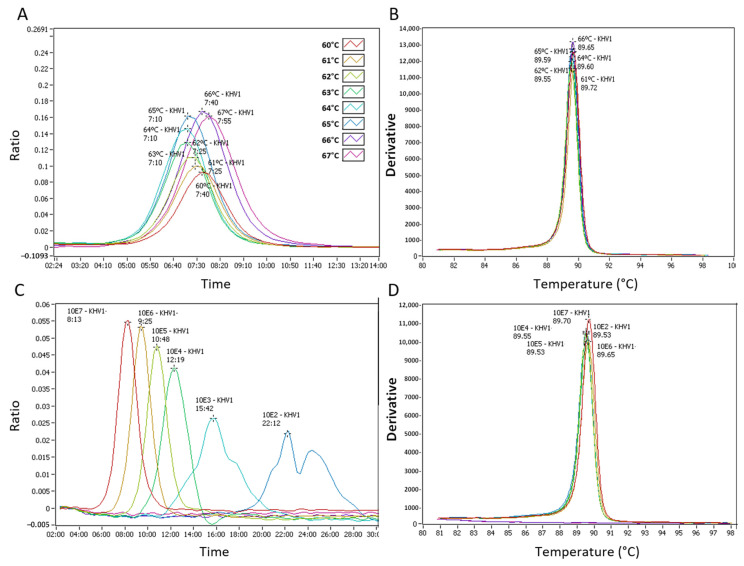
Cyprinid herpesvirus (CyHV)-3 loop-mediated isothermal amplification (LAMP) assay, test named KHV1. (**A**) Isothermal amplification of CyHV-3 DNA at different reactable temperatures from 60 °C to 67 °C. (**B**) Anneal derivative of the isothermal amplified products in the LAMP reaction showed in “A”. (**C**) Analytical sensitivity of CyHV-3 LAMP assay. The amplification graph shows serial dilutions from 10^7^ to 1 copy of a recombinant plasmid. (**D**) Anneal derivative of the isothermal amplified products in the LAMP reaction showed in “C”.

**Figure 3 animals-11-00459-f003:**
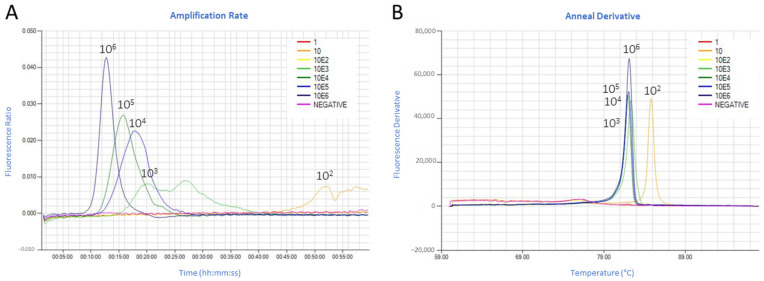
Carp edema virus (CEV) loop-mediated isothermal amplification (LAMP) assay. (**A**) Analytical sensitivity of CEV LAMP assay. The amplification graph shows serial dilutions from 10^6^ to 1 copy of a recombinant plasmid. (**B**) Anneal derivative of the isothermal amplified products in the LAMP reaction.

**Figure 4 animals-11-00459-f004:**
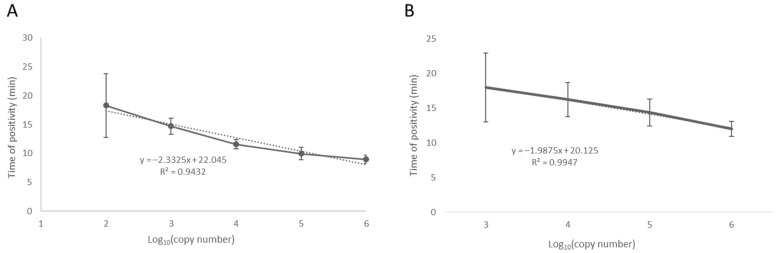
Linear correlation of (**A**) cyprinid herpesvirus (CyHV)-3 and (**B**) carp edema virus (CEV) plasmid copy number (expressed as Log_10_(x)) and the time of positivity (*Tp*).

**Figure 5 animals-11-00459-f005:**
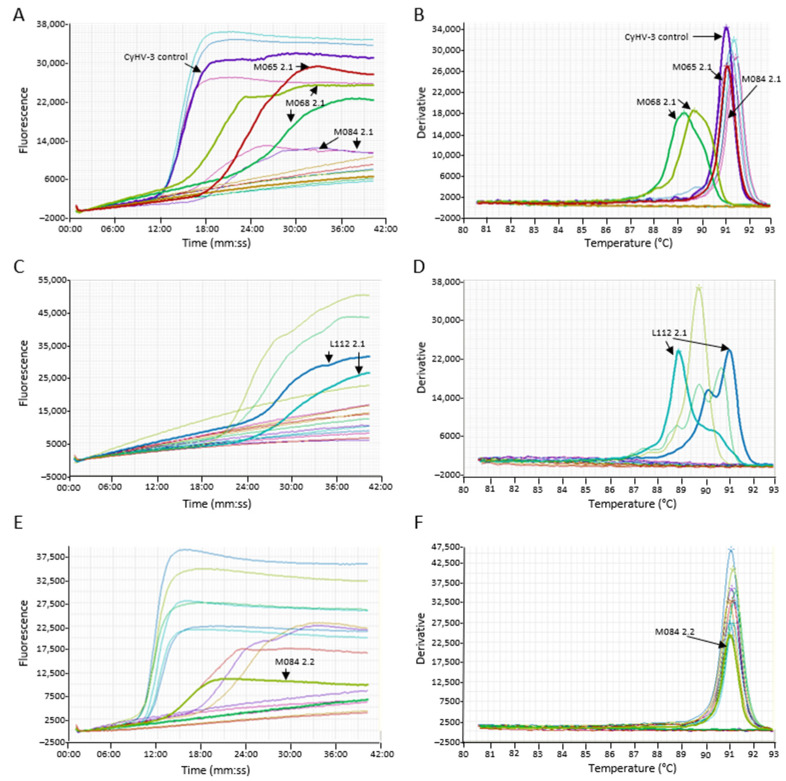
Examples of cross-reaction of the Cyprinid herpesvirus (CyHV)-3 loop-mediated isothermal amplification (LAMP) assay with other herpesviruses. (**A**,**C**,**E**) The amplification graph of archived DNA. (**B**,**D**,**F**) Anneal derivative of the LAMP products. Samples of interest are highlighted (arrows)—amplification of carp pox virus (CyHV)-1 samples: M065 2.1 and M068 2.1. Amplification of goldfish herpesvirus (CyHV-2) samples M084 2.1, M084 2.2, and L112 2.1.

**Figure 6 animals-11-00459-f006:**
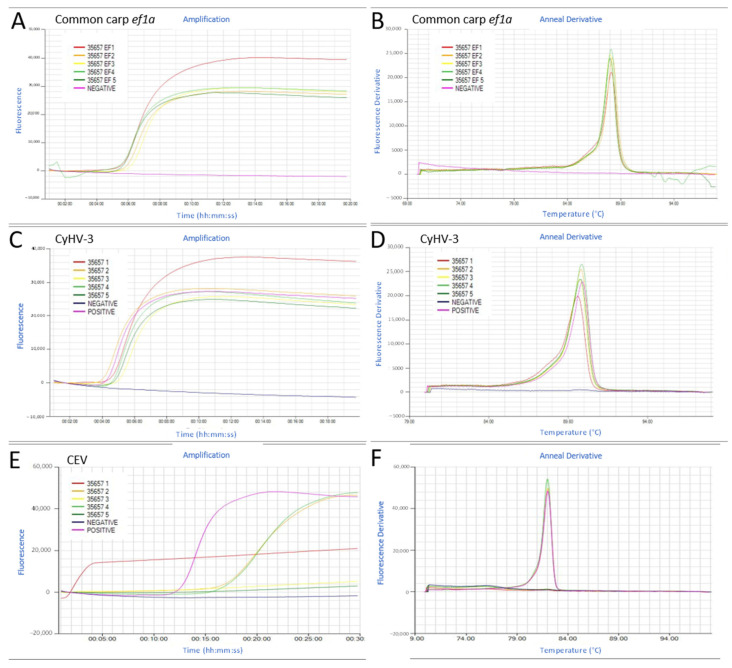
Examples of the detection of the common carp *elongation factor 1a* (*ef1a*) gene (**A**,**B**), the Cyprinid herpesvirus (CyHV)-3 (**C**,**D**), and the carp edema virus (CEV) (**E**,**F**) by loop-mediated isothermal amplification (LAMP) tests in common carp mucus swabs taken during disease investigations. (**A**,**C**,**E**) The amplification graph of tested swabs. (**B**,**D**,**F**) The anneal derivative graph of the corresponding LAMP products.

**Table 1 animals-11-00459-t001:** LAMP primer sequences designed for the Cyprinid herpesvirus-3 (CyHV-3) *orf43* gene (GenBank accession no. NC_009127), the Carp Edema Virus (CEV) *p4a* gene (GenBank no. MH397469.1), and the common carp *elongation factor 1 alpha* (*ef1a*) gene (GenBank no. AF485331.1). °C: Annealing temperature.

Assay	Primer	Sequence (5′-3′)	°C	Position
CyH-3 *orf43*	F3	CTGATGGTCCTGTCCGAG	64	72,147
	B3	ACTGGTTAATGCAGTCGTC		72,357
	FIP (F1c + F2)	CCGTGTTGAGGTAGAAGAGGCCCAGACCTACGTGAAAGC		-
	BIP (B1c + B2)	CTGGACCTTCCGTCTGCTGGTCTTGTTGCGGAAGGTG		-
	Loop-F	ACATGAGGCTGTTGTAGGTG		72,240
	Loop-B	GTACGAGTGCATCGCCTA		72,296
CEV *p4a*	F3	CCAGATTTGGGTTGAAACATG	64	139
	B3	GCAAAGTACTACCTCATCCAA		494
	FIP (F1c + F2)	CTTGCTCTAGTTCTAGGATTGTACTGTAACTCCTTGAGGAAT		-
	BIP (B1c + B2)	GTCTTGTCAAAGACAGACATCTGTTGACACAATTCCAGAACAAG		-
	Loop-F	ATTGCGGAATTCTAGATCAG		260
	Loop-B	GGATTCCTTTCCAGAACATAAC		387
Carp *ef1a*	F3	TGCCACTTCAGGATGTCTA	64	800
	B3	GGATGTCCTTGACAGACAC		1200
	FIP (F1c + F2)	GTCACAACCATACCAGGCTTGAATTGGAGGTATTGGAACTGTG		-
	BIP (B1c + B2)	GCCAACTTGACCACTGAGGTGTAGCCTCAGCAAGAGATTC		-
	Loop-F	GACACCAGTCTCCACACG		870
	Loop-B	AAGTCTGTTGAGATGCACCAT		965

**Table 2 animals-11-00459-t002:** Cyprinid herpesvirus (CyHV)-3 loop-mediated isothermal amplification (LAMP) results of 174 archived DNA samples extracted from gill tissues of various fish species. №: the number of samples tested for each group. CyHV-1: carp pox virus; CyHV-2: herpesviral hematopoietic necrosis virus; AngHV: eel herpesvirus; HV: uncharacterized herpesvirus. MT: LAMP product melting temperature. NA: no applicable. LAMP interpretation column shows the number of LAMP false positives and false negatives per group.

Pathogen	№	LAMP Amplification	MT	LAMP Interpretation	
				False-Positive	False-Negative
CyHV-3	85	76	Correct	NA	9 (˂LOD)
CyHV-3 variant	30	0	NA	0	NA
CyHV-1	28	3	Different in 2 out of 3	1	NA
CyHV-2	10	3	Same as CyHV3, coinfection confirmed in one sample	2	NA
AngHV	2	1	Different	0	NA
HV	2	1	Same as CyHV-3	1	NA
Negative samples	47	0	NA	0	NA

**Table 3 animals-11-00459-t003:** Evaluation of the point of care (POC) loop-mediated isothermal amplification (LAMP) test in mucus swabs of common carp taken during disease investigations. №: number of swabs analyzed per site. “LAMP POCT” columns show the percentage (%) of swabs per site that gave a positive result in either the common carp *elongation factor 1 alpha* (*ef1a*), the Cyprinid Herpesvirus (CyHV)-3, or the Carp Edema Virus (CEV) LAMP assay. “Site designation” shows if a site was considered positive (+) or negative (-).

Site	№	LAMP POCT (%)			Site Designation			
		*ef1a*	CyHV-3	CEV	LAMP POCT		Statutory Diagnostics	
					CyHV-3	CEV	CyHV-3	CEV
A	5	80	100	0	+	-	+	-
B	1	100	100	0	+	-	+	-
C	5	100	100	40	+	+	+	+
D	1	100	100	0	+	-	+	-
E	5	100	80	0	+	-	+	-
F	1	100	100	0	+	-	+	-
G	5	0	20	0	+	-	+	-
H	1	100	0	0	-	-	-	-
I	4	100	100	0	+	-	+	-
J	2	100	100	0	+	-	+	-
K	10	40	70	0	+	-	+	-
L	10	20	100	0	+	-	+	-
M	2	50	0	0	-	-	-	-
N	3	100	66	0	+	-	+	-
O	3	100	0	0	-	-	-	-
P	5	40	100	0	+	-	+	-

**Table 4 animals-11-00459-t004:** Comparison of fast diagnostic assays with potential as a point of care test for detecting Cyprinid herpesvirus (CyHV)-3 or carp edema virus (CEV). LAMP: loop-mediated isothermal amplification; RPA: Recombinase Polymerase Amplification. AuNPs: gold nanoparticles. Time: shows the time in minutes for the test completion and its visualization (it does not include the time for DNA extraction). LOD: limit of detection. *tk*: *thymidine kinase* gene. *mpc*: major capsid protein. *p4a*: partial core protein 4a. LFD: lateral flow device. *n* = number of samples. Specificity refers to tests conducted with relative herpesviruses.

Assay	Test Visualization	Time	Target	LOD	Specificity	Sample Type	Internal Control	Reference
LAMP	Gel electrophoresis	˃60	CyHV-3 *tk*	10^−6^ stock dilution	Only *in silico* tested	Homogenates	None	[38]
LAMP	Turbidity	60	CyHV-3	10^−7^ stock dilution; 6 copies	No cross-reaction with fish HVs (*n* = 3)	Homogenates	None	[41,42]
RPA	LFD	˂30	CyHV-3 *mcp*	10 copies	No cross-reaction with CyHV-2 (*n* = 1)	White blood cells	None	[40]
Multiplex RPA	LFD	25	CyHV-3 *tk* CEV *p4a*	CyHV-3: 21 copies CEV: 1.8 copies	No cross-reaction with CyHV-1 or CyHV-2 (*n* = 2)	Homogenates	None	[31]
AuNPs hybridization	Colorimetric	21	CyHV-3 *tk*	150 copies	No cross-reaction with CyHV-1 or CyHV-2 (*n* = 2)	Homogenates	None	[39]
LAMP	Fluorescence real-time	˂20	CyHV-3 *orf43*	100 copies	4.4% cross-reaction with other fish HVs (*n* = 72)	Homogenates Gill/skin swabs	Host *ef1a*	This study
LAMP	Fluorescence real-time	˂20	CEV *p4a*	1000 copies	No cross-reaction with fish HVs (*n* = 3)	Homogenates Gill/skin swabs	Host *ef1a*	This study

## Data Availability

The data presented in this study is contained within this article or its Appendix A.

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
