# Peer review of "A Seasonal Study of Koi Herpesvirus and Koi Sleepy Disease Outbreaks in the United Kingdom in 2018 Using a Pond-Side Test"

_animals, 2021, doi:10.3390/ani11020459_

Round 1

Reviewer 1 Report

The topic of this study is innovative and interesting, where the researchers studied the potency of a "pond-side test (LAMP assays)" for rapid and on-site identification of DNA viruses [Cyprinid herpesvirus (CyHV)-3 and carp edema virus (CEV)], which are the causative agents of koi and common carp diseases.

The experiments are well designed and performed. The paper is well written.

The introduction is ok and properly reviewed the relevant literature.

The methods used to perform the study are clear and adequate.

The results presented are adequate.

The tables and figures used to show them are adequate.

The discussion is fairly comprehensive.

The conclusion was supported by the results and clearly expressed the main hypothesis of the study. 

Therefore, I think this paper is suitable to publish in the "Animals" after addressing the minor points:

Line 47-49: Add reference.

Line 57-59: Revise the sentence.

Line 359-367: This paragraph should be removed from the results section and added to the material and method section.

The discussion section is repetitive when compared to the results section. This duplication (results) should be eliminated.

Author Response

Dear reviewer, thank you for your kind comments and for your time in reviewing this manuscript. We have amended those lines, added the missing reference, and revised the discussion to avoid repetition of the results.

Kind regards

Authors

Reviewer 2 Report

You've done a great job! It was really interesting. But, on the other hand, such methodical paper should be, in my opinion, shorter. Some sections of the text could be omitted.

Author Response

Dear reviewer, thank you for your kind comments and for your time in reviewing this manuscript. We have edited the manuscript to avoid repetitions and shortened the discussion. We hope that the readability of the text has improved.

Kind regards

Authors

Reviewer 3 Report

the manuscript described the use of LAMP assays to detect the orf43 gene of CyHV-3 European genotype, the p4a gene of the CEV genogroup I, and the ef1a gene of the host common carp, as an internal control. Although the few and minimal limitations shown by the assay (LOD, specificty, and sensitivity), the overall good performance was good. the assay is easy of use, cost-effective, and quick.

The authors have presented their work properly and have discussed all obtained results in a proper way. I believe that the authors will keep working to improve the assay; especially in terms of using proper internal control and to solve the main issue with cross reactivity.

Author Response

Dear reviewer, thank you for your kind comments and for your time in reviewing this manuscript. We will keep working on improving this test and hope to conduct a ring test including this and other published tests which will allow for its international validation and inclusion on contingency plans.

Kind regards

Authors